# Dietary Zinc Supplemented in Organic Form Affects the Expression of Inflammatory Molecules in Swine Intestine

**DOI:** 10.3390/ani13152519

**Published:** 2023-08-04

**Authors:** Ramya Lekha Medida, Ashok Kumar Sharma, Yue Guo, Lee J. Johnston, Pedro E. Urriola, Andres Gomez, Milena Saqui-Salces

**Affiliations:** 1Department of Animal Science, University of Minnesota, Saint Paul, MN 55108, USA; medid002@umn.edu (R.L.M.); ashoks773@gmail.com (A.K.S.); guoxx390@umn.edu (Y.G.); urrio001@umn.edu (P.E.U.); gomeza@umn.edu (A.G.); 2West Central Research and Outreach Center (WCROC), University of Minnesota, Morris, MN 56267, USA; johnstlj@umn.edu

**Keywords:** organic zinc, zinc sources, immune response, organoids

## Abstract

**Simple Summary:**

Zinc is an essential mineral supplemented to human and pig diets as inorganic (inorganic salts) or organic (bound to an organic compound) sources. Some studies reported that pigs fed diets with organic zinc sources respond better to stress conditions than those fed diets with inorganic sources. In this study, we identified genes in the small intestine that may explain those differences. We fed finisher pigs with diets containing organic or inorganic zinc sources for 32 days and conducted gene expression analysis of intestinal ileum samples. The organic Zn source induced lesser level of pro-inflammatory cytokine *interleukin 18* (*IL18*), but greater *Toll-like receptor 2* (*TLR2*) gene expression than inorganic Zn, suggesting that an organic Zn source may improve the condition and immune response of animals under stress by modulating gene expression in the intestine. We also explored if intestinal epithelial cells, which are in direct contact with diet components, are the main cells showing these changes. We exposed intestinal organoids or epithelial “mini-guts” to zinc sources in vitro. Organoids showed changes related to Zn supplementations but did not reproduce the changes of immune-related genes observed in the animal samples, indicating cells other than epithelial are affected by Zn sources.

**Abstract:**

Animals receiving Zinc (Zn) dietary supplementation with organic sources respond better to stress than inorganic Zn sources supplementation. The study aimed to identify the effect of different Zn sources on intestinal epithelial gene expression. In total, 45 pigs (9 per treatment) (77.5 ± 2.5 kg weight) were fed for 32 days, a corn-soybean meal diet without supplemented Zn (ZnR) or supplemented with 50 and 100 ppm of inorganic ZnCl_2_ (Zn50 and Zn100), and amino acid-bound organic Zn sources (LQ50 and LQ100). Gene expression changes form RNA-seq in ileum tissues of ZnR revealed changes associated with Zn insufficiency. Comparing organic with inorganic Zn sources by one-way ANOVA, pro-inflammatory cytokine *interleukin 18* (*IL18*) was downregulated (*p* = 0.03) and *Toll-like receptor 2* (*TLR2*) upregulated (*p* = 0.02). To determine the role of epithelial cells in response to dietary Zn, swine intestinal organoids (enteroids) were exposed to Zn restriction, ZnCl_2_ or LQ-Zn. In enteroids, *ZIP4* expression decreased with added Zn compared with Zn-restriction (*p* = 0.006) but Zn sources did not affect (*p >* 0.05) *IL18* or *TLR2* expression. These results suggest that organic Zn may stimulate TLR2 signaling possibly affecting immune response, while decreasing the proinflammatory cytokine *IL18* expression in non-epithelial cells of intestinal mucosa.

## 1. Introduction

Zinc (Zn) is an essential mineral required by mammals for growth, development, and overall health [1,2,3,4]. The National Research Council (NRC) [5] recommends that diets contain 50–100 mg Zn/kg diet to meet the requirements for normal growth and development of pigs. Zinc is supplemented in organic (Zn bound to organic molecules) or inorganic sources (Zn from inorganic salts) to swine diets. Reports indicate that Zn supplementation with organic sources has advantages over inorganic Zn supplementation in different species including swine, chickens, cattle, and rodents [6,7,8,9,10,11,12,13,14,15,16]. In growing pigs, heat stress reduction of growth performance and intestinal permeability was alleviated by feeding 60–200 ppm of organic Zn [9,10,11,12]. Sows fed diets supplemented with 45 ppm of organic Zn in combination with 80 ppm of inorganic Zn had fewer hoof lesions than sows fed diets supplemented exclusively with 125 ppm of inorganic Zn from Zn oxide [8]. Despite extensive studies focused on organic Zn in animal performance, understanding of the mechanisms by which organic and inorganic Zn sources differ in their physiological functions is incomplete.

Researchers often assumed that differences in physiological responses observed in animals fed diets with organic and inorganic Zn supplementation may be due to the differences in Zn bioavailability; however, there are some discrepancies in the literature on this issue. Some studies demonstrated an increase in bioavailability of Zn supplemented by organic sources to pig diets [17], others have claimed either no difference [18,19] or an increased Zn bioavailability from inorganic sources [20]. Therefore, it is possible that, in addition to differences in bioavailability, tissues respond differently to organic and inorganic sources of Zn.

The small intestine is the main site for Zn acquisition, with the intestinal epithelium being responsible for the uptake and transport of Zn from the diet. The objective of the present study was to determine if dietary organic and inorganic Zn sources elicit differential responses in the small intestine. Here, we used RNA sequencing (RNA-seq) to identify genes and signaling pathways changing in the intestine of pigs fed diets supplemented with organic or inorganic Zn sources. Further, we sought to determine if the intestinal epithelium, being the layer directly exposed to different Zn sources in the diet, demonstrates the changes observed in the animal sample analysis. We focused our analysis on the differentially expressed genes, interleukin 18 (*IL18*) and Toll-like receptor 2 (*TLR2*), which are of interest because of their role in inflammatory responses, and on Zn transporters *ZIP4* and metallothionein (*MT1*) for proof of concept. To that end, we evaluated the gene expression changes of *ZIP4*, *MT1*, *IL18*, and *TLR2* in swine enteroids exposed to organic and inorganic sources of Zn.

## 2. Materials and Methods

### 2.1. In Vivo Study

Animal care and experimental protocols were approved by the Institutional Animal Care and Use Committee of the University of Minnesota, protocol: #1709-35112A. Experimental procedures were carried out at the University of Minnesota West Central Research and Outreach Center (WCROC) in Morris, MN. The experiment was conducted with finisher pigs (*n* = 45; 21 males and 24 females; Landrace/Yorkshire crossbreed; body weight = 77.5 ± 2.5 kg, age = 17–18 weeks). Prior to the experiment, the pigs were provided with diets containing adequate levels of zinc (Zn). Pigs were housed in pens with unrestricted access to both feed and water for the entire 32-day duration of the study. The pigs were divided into five groups, each assigned to a specific dietary treatment, with nine pigs in each group. Weekly records were taken of the body weight (BW) of the pigs and the amount of feed consumed per pen.

Except for zinc (Zn), the experimental diets were carefully formulated to meet or exceed the recommended nutritional requirements [5] for finisher pigs. The basal diet comprised corn, soybean meal (46% protein), and soybean oil (Table 1). To create the experimental diets, a Zn-free vitamin-mineral mix (Table 2) was added and supplemented with either 50 ppm or 100 ppm of Zn from two different sources: ZnCl2 (inorganic source) or LQ-Zn (amino acid-Zn complex, organic source, Zinpro Corporation, Eden Prairie, MN, USA), following previously established protocols [21]. The Zn-free vitamin–mineral mix was sourced from Agri-Nutrition Inc. (Shakopee, MN, USA), while Zn products were provided by Zinpro Corporation (Eden Prairie, MN, USA). All other ingredients used in the diets were locally sourced.

Feed samples underwent analysis for total zinc (Zn) concentration at the Soil Testing Laboratory, University of Minnesota, using inductively coupled plasma optical emission spectrometry (ICP-OES). The results revealed the following Zn concentrations in the diets: 34 ppm for ZnR (no Zn supplementation), 88 ppm for Zn50 (50 ppm Zn from ZnCl2), 147 ppm for Zn100 (100 ppm Zn from ZnCl2), 87 ppm for LQ50 (50 ppm Zn from amino acid–Zn complex), and 134 ppm for LQ100 (100 ppm Zn from amino acid–Zn complex).

#### 2.1.1. Sample Collection

On day 32 of the experiment, blood samples were obtained and left to clot for a minimum of 20 min. Subsequently, they were centrifuged at 2000× *g* for 10 min in a refrigerated centrifuge operating at 4 °C. The resulting serum samples were collected and stored at −80 °C until analysis. To measure serum zinc (Zn) levels, inductively coupled plasma-mass spectrometry (ICP-MS) was utilized at the Michigan State University Veterinary Diagnostic Laboratory.

At the end of experiment, all pigs were humanely euthanized using a captive bolt followed by exsanguination. Immediately after euthanasia, tissue samples were collected from the jejunal (1 m distal to pyloric sphincter) and ileal (15 cm proximal to ileocecal valve) regions of the intestine. These tissue samples comprised epithelial, stromal, muscle, and immune cells. The collected samples were promptly snap-frozen in liquid nitrogen and then stored at −80 °C until further processing for analysis. For RNA sequencing, the ileal samples were utilized, while both ileal and jejunal samples were employed for gene expression analysis through quantitative PCR.

#### 2.1.2. RNA Sequencing and Differential Expression Analysis

Jejunal and ileal RNA samples were extracted using the Qiagen RNeasy Mini Kit (Qiagen, Germantown, MD, USA) following manufacturer’s instructions. Jejunal RNA samples were used for qPCR analysis and ileal RNA samples were used for both qPCR analysis and RNA sequencing; however, five ileal samples were removed from the qPCR and RNA sequencing analysis due to low RNA quality not suitable for sequencing (two samples from ZnR group, and one sample each from Zn50, Zn100 and LQ100 groups). Sequencing and cDNA library construction from the RNA samples was completed at the University of Minnesota Genomic Center. RNA library preparation was carried out using TrueSeq stranded RNA kit and sequencing was run on the Illumina HiSeq 2500 platform. Quality control, trimming and alignment of reads, differential expression analysis, and pathway analysis was completed as described elsewhere [21]. Briefly, low-quality bases were trimmed using Trimmomatic [22] and trimmed paired end sequences were mapped to reference Suscrofa 11.1 (Enseml) using Kallisto [23]. Transcripts per million for each gene were generated and differential gene expression analysis was performed using DESeq2 package in R [24,25]. Gene symbols and fold change values for significantly different genes were submitted to PANTHER [26] and Ingenuity Pathway Analysis (IPA; Ingenuity^®^ Systems, http://www.ingenuity.com (accessed on 31 March 2021) to identify functional classification of the DE genes, significantly enriched canonical pathways and network interactions among the DE genes. Raw data and processed data have been deposited in the NCBI Gene Expression Omnibus (GEO) under accession number GSE181343.

#### 2.1.3. Quantitative PCR (qPCR) for Gene Expression Analysis

For cDNA synthesis, 500 ng of total RNA extracted from ileal and jejunal samples (*n* = 9 per group) was converted using the High-Capacity cDNA Reverse Transcription Kit (Applied Biosystems, Foster City, CA, USA). The qPCR reactions were prepared with 100 nM of forward and reverse primers and 10 ng of cDNA template, utilizing the PowerUp SYBR Green PCR Master Mix (ThermoFisher Scientific, Waltham, MA, USA). The qPCR conditions consisted of an initial activation at 95 °C for 10 min, followed by 40 cycles of denaturation at 95 °C for 15 s and annealing at 60 °C for 60 s. Glycerahdehyde-3-phosphate dehydrogenase (*GAPDH*) was selected as the reference gene [27,28,29], after ensuring that samples exhibited no more than one cycle difference in the *GAPDH* amplification [30]. The primer sequences for the chosen genes are provided in Table 3. To determine gene expression changes, the 2^−ΔΔCt^ method was employed [31].

### 2.2. In Vitro Study

#### 2.2.1. Swine Enteroids

Swine enteroids were cultured, passaged, and plated on 6.5 mm transwell inserts (Corning, Ref: 3470), as described previously [32]. Cell growth was monitored until the membrane was covered by cells. Transepithelial resistance (TER) was measured with a Millicell-ERS2 Volt-Ohm meter (Millipore Sigma, St. Louis, MO, USA) before media change and the experiment was carried out when TER reached a minimum of 600 ohms/cm^2^ and was constant. Swine enteroids were maintained in Intesticult Organoid Growth Medium (Human; Stemcell Technologies, Vancouver, BC, Canada) which will be referred to as the culture media in this report.

#### 2.2.2. Zn-Restriction of Swine Enteroids Pilot

Because the culture media contained Zn, a pilot study was performed to define the culture conditions necessary for swine enteroids to present changes associated with Zn restriction by evaluating the expression of *ZIP4* and *MT1.* On the day of the pilot experiment, culture media was replaced by fresh culture media in the bottom wells (basal side) and by porcine Ringers’ solution (130 mM sodium chloride, 6 mM potassium chloride, 0.7 mM magnesium chloride, 1.5 mM calcium chloride, 19.6 mM sodium bicarbonate, 0.29 mM monosodium phosphate, 1.3 mM monosodium phosphate and 10 mM glucose) in the transwell inserts (luminal side) and incubated for 2 h at 37 °C and 5% CO_2_. After 2 h, the solution in the transwell inserts was replaced with 100 μL of culture media (CM) or the vehicle Porcine Ringers’ (VEH). Enteroids were collected at the start of the treatments (0 h) and 3 h after treatment. Total RNA was extracted, 500 ng of extracted RNA was used for reverse transcription to cDNA and qPCRs were performed as described previously. Fold change values were calculated by normalizing the gene expression in enteroids treated with CM and VEH to enteroids at 0 h set to 1. This experiment was repeated three times.

#### 2.2.3. Swine Enteroids Treatments

After confirmation that the VEH compared with the CM induced changes in the expression of *ZIP4* similar to Zn restriction, the following experiment was performed to evaluate if the changes in expression of *IL18* and *TLR2* are specific to the intestinal epithelium. On the day of experiment, media was replaced by fresh culture media in the bottom wells (basal side) and by porcine Ringers’ solution in the transwell inserts (luminal side) and incubated for 2 h at 37 °C and 5% CO_2_. After 2 h, the solution in the transwell inserts was replaced with 100 μL of the following: (a) the vehicle Porcine Ringers’ (VEH); (b) VEH + 50 ppm ZnCl_2_; (c) VEH + 100 ppm ZnCl_2_; (d) VEH + 50 ppm LQ-Zn; or (e) VEH + 100 ppm LQ-Zn. Enteroids were collected 3 h after treatment. Total RNA was extracted, 500 ng of extracted RNA was used for reverse transcription to cDNA and qPCRs were run as described previously. This experiment was repeated three times.

## 3. Data Analyses

The changes in weight data of the pigs were subjected to analysis using the GLM procedure of SAS (SAS Inst. Inc., Cary, NC, USA), with individual pigs considered as the experimental units. The fixed effects taken into consideration were diets and sex, while initial body weight (BW) was considered as a covariate. Multiple comparisons of means were conducted using the Student–Newman–Keuls (SNK) method. The results are presented as least squares means, and treatment effects were deemed significant if *p* ≤ 0.05.

For the comparisons of serum Zn levels and gene expression among treatments, a one-way ANOVA with Kruskal–Wallis multiple comparisons test was employed using GraphPad Prism 8.1 software for Windows (GraphPad Software, Boston, MA, USA). The data are expressed as means with standard error (±SEM), and significance was considered at *p* ≤ 0.05.

## 4. Results

### 4.1. Animal Growth

Average daily weight gain (*p =* 0.20) and average daily feed intake on a pen basis were not different among pigs fed the dietary treatments (Appendix A).

### 4.2. Serum Zinc Levels

Compared with pigs fed the ZnR diet, pigs in other treatment groups had greater serum Zn concentration (*p <* 0.0001). The serum concentration of Zn in pigs fed LQ50 diet was less than that of pigs fed LQ100 (*p* = 0.032), but no differences were observed between the serum Zn concentration of pigs supplemented with Zn50 and Zn100 (*p* = 0.39; Figure 1).

### 4.3. Differentially Expressed Genes (DEGs)

The DEGs analysis revealed gene changes associated with zinc restriction when ileal samples from the ZnR group were compared with all other groups. Differences identified among ZnR and Zn supplemented groups were reported elsewhere [21] and the focus of the present report is on differences between Zn sources. A total of 291 DEGs were observed in the ileum of pigs supplemented with Zn50 compared with the pigs supplemented with LQ50, of which 186 genes were downregulated and 105 genes were upregulated. Of the 369 DEGs observed in the ileum of pigs supplemented with Zn100 compared with those supplemented with LQ100, 213 genes were downregulated, and 156 genes were upregulated. Gene names and fold change values of DEGs are provided in Appendix A.

Functional classification of the DEGs identified using PANTHER [26] revealed active genes in the ileum of pigs fed the Zn50 supplemented diet compared with those fed the LQ50 diet. Those genes were related to structural molecular activity (2 genes), transcription regulator activity (8 genes), transporter activity (11 genes), binding protein (46 genes), catalytic activity (46 genes), molecular function regulators (9 genes), and molecular transducer activity (3 genes) (Appendix A). Classification of DEGs found when comparing samples from pigs fed the Zn100 diet with samples from pigs fed the LQ100 diet were related to structural molecular activity (20 genes), transcription regulator activity (11 genes), translation regulator activity (2 genes), transporter activity (7 genes), binding protein (63 genes), catalytic activity (41 genes), molecular function regulators (4 genes) and molecular transducer activity (3 genes) (Appendix A).

Pathway analysis of DEGs using IPA identified 22 pathways significantly regulated between samples from pigs fed diets supplemented by Zn50 and LQ50 (Table 4) and 37 pathways significantly regulated between pigs fed Zn100 and LQ100 diets (Table 5). Based on z-scores, we identified downregulated pathways when the Zn50 group was compared with LQ50 group. These pathways are related to neuroinflammation signaling, Fcγ receptor-mediated phagocytosis, paxillin signaling, colorectal cancer metastasis signaling, and signaling by Rho family GTPases (Figure 2A). The top five upregulated pathways when Zn50 group was compared with LQ50 group were Cholecystokinin/Gastrin-mediated signaling, RhoGDI signaling, acute phase response signaling, PI3K signaling in B lymphocytes, and NFкB signaling. The comparison between ilea of pigs supplemented by Zn100 and LQ100 yielded downregulation of EIF2 signaling, production of nitric oxide and reactive oxygen species in macrophages, nuclear factor kappa B (NFкB) signaling, Tec kinase signaling, and signaling by Rho family GTPases. The upregulated pathways in the Zn100 group compared with LQ100 group were p70S6K signaling, NRF2-mediated oxidative stress response, ErbB4 signaling, IL-3 signaling, and paxillin signaling (Figure 2B). Gene names and their changes are presented in Appendix A.

When ilea of pigs fed with the Zn50diet were compared with that of pigs fed the LQ50 diet, the top network identified included events related to carbohydrate metabolism and small molecule biochemistry, with a central node identified as tripartite motif containing-25 (TRIM-25) (*p* ≤ 0.05, Figure 3A). The top network identified for the comparison between Zn100 and LQ100 included events related to protein synthesis, and RNA damage and repair with multiple network nodes (*p* ≤ 0.05, Figure 3B).

Because of the reported effects of feeding diets supplemented with organic and inorganic sources of Zn on animal’s response to stress, we focused the subsequent analysis on genes associated with immune responses. From the DEGs (Appendix A) and pathway analyses, we identified that interleukin 18 (*IL18*) was upregulated (Log2 fold change: 0.53; *p* = 0.03) and Toll-like receptor 2 (*TLR2*) was downregulated (Log2 fold change: −2.5; *p* = 0.02) in the ileum of pigs fed the Zn50 diet compared with those fed the LQ50 diet.

To further validate the RNAseq findings, the expression of Zn transporter *ZIP4*, Zn cellular transport/storage gene metallothionein 1 (*MT1*) and immune response molecules *IL18* and *TLR2* were evaluated in ileal and jejunal samples using qPCR. The increase in expression of *MT1* and decrease in expression of *ZIP4* in response to Zn supplementation has been well established in previous studies [33,34,35,36]. In the jejunum of pigs supplemented by Zn100 and LQ100, we confirmed decreased expression of *ZIP4* (Figure 4A) compared with ZnR group. In LQ100, Zn50 and LQ100 groups, we observed increased expression of *MT1* (Figure 4B) compared with pigs fed ZnR diets. However, the gene expression of *ZIP4* and *MT1* between organic and inorganic Zn supplementation was not different in the jejunum. The expression of *IL18* among jejunal samples of pigs supplemented with different Zn sources was not different. However, compared with the samples from the ZnR diet group, *IL18* expression increased in Zn100 (*p =* 0.048) and was not different from that observed in the LQ100 group (Figure 4C). The expression of *TLR2* in the jejunum of pigs supplemented with LQ50 was lower than that seen in pigs fed the ZnR (*p =* 0.008) and LQ100 (*p =* 0.0008) diets (Figure 4D).

In ileal tissues, the expression of *ZIP4* (Figure 5A) decreased and expression of *MT1* (Figure 5B) increased in response to Zn supplementation when compared with feeding ZnR diets. In addition, the source of Zn did not change the expression of *ZIP4* or *MT1.* Ileal expression of *IL18* was increased in the Zn100 group compared with ZnR (*p =* 0.0022), Zn50 (*p =* 0.0003) and LQ100 (*p =* 0.004) (Figure 5C). The expression of *TLR2* increased in pigs fed Zn50 (*p =* 0.0002) and LQ50 (*p =* 0.0037), compared with the ileum of ZnR fed pigs. Also, the expression of *TLR2* decreased in ileum of pigs fed Zn100 (*p =* 0.001) and LQ100 (*p =* 0.019) compared with those fed the Zn50 and LQ50 diets (Figure 5D).

### 4.4. In Vitro Study

The intestinal samples collected from animals contained epithelial, stromal, muscle and immune cells. Thus, we sought to determine if the intestinal epithelium, being the layer directly exposed to different Zn sources in the diet, demonstrates the changes observed in the animal sample analysis. To that end, we evaluated the gene expression changes of *ZIP4*, *MT1*, *IL18*, and *TLR2* in swine enteroids exposed to organic and inorganic sources of Zn. We first restricted Zn supplementation in the enteroid model; this was done to define the time and conditions necessary for swine enteroids to exhibit characteristic gene expression changes associated with Zn deficiency. In this way, we demonstrated the control conditions that would define the enteroids’ response to Zn supplementation. After 3 h of Zn depravation in the luminal side, the Zn restricted enteroids (VEH) showed greater (*p =* 0.006) expression of *ZIP4* compared with the enteroids exposed to luminal culture media (Figure 6A), thereby confirming that swine enteroids do respond to Zn restriction. However, no difference was observed in the expression of *MT1* (Figure 6B), probably due to presence of Zn from culture media on the basal side.

To determine if the changes in *IL18* and *TLR2* expression observed in pigs’ tissues occur in the epithelial cells, we treated swine enteroids with different Zn sources. Although we observed numerical differences, no significant changes were observed in the expression of *IL18* and *TLR2* or in the expression of *ZIP4* and *MT1* in swine enteroids treated with VEH, Zn50, Zn100, LQ50, or LQ100 (Figure 7).

## 5. Discussion

The current study used an exploratory approach to understand the differential responses of the intestine to organic and inorganic dietary Zn sources. Growth performance of pigs can be affected by multiple factors including stress, inflammation, and environmental conditions [37,38,39,40]. Since no clinical signs of deficiency or differences in weight were observed in the present study, we suggest that any changes in gene expression can be attributed to the different Zn sources. Although there are some discrepancies among previous studies about the bioavailability of Zn from different sources, the lack of differences in serum Zn concentration between pigs supplemented with organic or inorganic Zn observed in this study is in agreement with other studies in pigs that showed no differences in serum Zn concentrations [18,19]. Further, the serum Zn levels observed in supplemented pigs match levels considered adequate for pigs [41,42,43], while the ZnR group showed Zn serum concentration below those levels, suggesting that the duration of the experiment was enough to identify early changes in response to Zn restriction.

The decrease in expression of *ZIP4* and increase in expression of *MT1* in the ileum of Zn supplemented pigs compared with those in the ZnR diet, is proof of concept that, under the conditions of this study, the pig intestine responded to Zn supplementation, as demonstrated in other studies [33,34,35,36]. We further demonstrated that supplementation by organic and inorganic sources of Zn did not result in differential expression of the Zn transporters *ZIP4* or *MT1* in pig intestines. Similar observations regarding these two genes were reported in a previous study in which pigs were supplemented with 110 ppm of organic Zn compared with pigs supplemented 110 ppm inorganic Zn [44]. We previously explored the changes associated with Zn supplementation (in contrast to Zn restriction) in the swine ileum and discussed gene expression and pathway changes in detail [21].

In the present study, RNA-seq analysis comparing Zn sources revealed differences in immune response-related pathways. Specifically, we identified changes in genes associated with TRIM-25, as indicated by the network analysis. TRIM-25 plays a crucial role in various immune-related processes, including innate immune response, interferon-gamma mediated signaling pathway, positive regulation of NFкB cascade, and defense response to viral entry [45].

Among the immune-related genes, *TLR2* and *IL18,* emerged consistently in the differentially regulated pathways and were selected as our primary focus for confirming the RNA-seq findings. Gene expression analyses from RNA-seq demonstrated that *TLR2* was downregulated while *IL18* was upregulated in the intestine of pigs supplemented with inorganic Zn, in comparison to those supplemented with the organic source. These findings suggest that organic Zn might modulate *TLR2* signaling, leading to reduced production of the proinflammatory cytokine *IL18* and potentially enhancing epithelial barrier function.

To further comprehend the mechanism of action of organic Zn sources, future studies are needed to explore changes of other immune-related genes identified in the RNA-seq analysis, and changes at the protein level. These investigations would provide deeper insights into the immunomodulatory effects of organic Zn and its potential benefits.

Expression of *IL18* was reduced in samples from pigs fed LQ100 compared with those fed Zn100. This observation may indicate that Zn source is relevant for enhancing stress responses [9,10,11,12] with organic Zn potentially decreasing the production of the proinflammatory cytokine IL-18, an important mediator of intestinal inflammation [46]. In line with these observations, a previous study in chickens fed organic Zn and challenged by coccidia and clostridium bacterial infection showed increased expression of *TLR2* and decreased expression of the proinflammatory cytokine *IL-8* in the jejunum, compared with birds fed diets supplemented with inorganic Zn [47]. To our knowledge, our study is the first to report changes in *TLR2* and *IL18* expression in pigs in response to different Zn sources. This differential gene response in the intestine of pigs, depending on organic or inorganic sources of Zn requires further investigation.

The intestinal epithelium modulates immune responses by directly modifying the expression of immune-related molecules or by relaying signals that, in turn, activate immune cells in the intestinal mucosa. We used enteroids to determine if Zn sources affect the expression of *TLR2* and *IL8* in the intestinal epithelium. We observed that expression of *MT1* and *ZIP4* in response to Zn supplementation in swine enteroids was in a manner consistent with the reports in the literature, thereby supporting their use in future studies to understand epithelial signaling pathways affected by Zn sources; however, we did not observe changes in *TLR2* or *IL8* expression. One interpretation for this lack of differences in swine enteroids exposed to Zn sources is that Zn effects on these genes may be stronger in non-epithelial cells. Other potential explanations are that the stress-free environment of in vitro culture, the presence of Zn in the basal compartment of the transwell, and the variability intrinsic to the culture system within experimental replicates may have limited our capacity to detect changes in gene expression. Further research is necessary to evaluate how much Zn is mobilized from the basal membrane of intestinal epithelial cells if there is luminal deficiency of Zn. Also, it will be important for future in vitro studies to involve an infection challenge; this procedure may fully define the effect of Zn sources when intestinal epithelial cells are challenged with stress conditions.

## 6. Conclusions

The present study identified genes and pathways that are modulated by dietary Zn sources in pigs. The pathway analysis suggested that organic and inorganic Zn sources affect pathways related to immune response, in particular those involving *IL18* and *TLR2*. We demonstrated, using swine enteroids, that intestinal epithelial cells are responsive to Zn supplementation and the intestinal response to Zn sources requires the participation of other cell types. More research is necessary to define the how Zn sources influence the intestinal mucosa during stress and inflammation.

## Figures and Tables

**Figure 1 animals-13-02519-f001:**
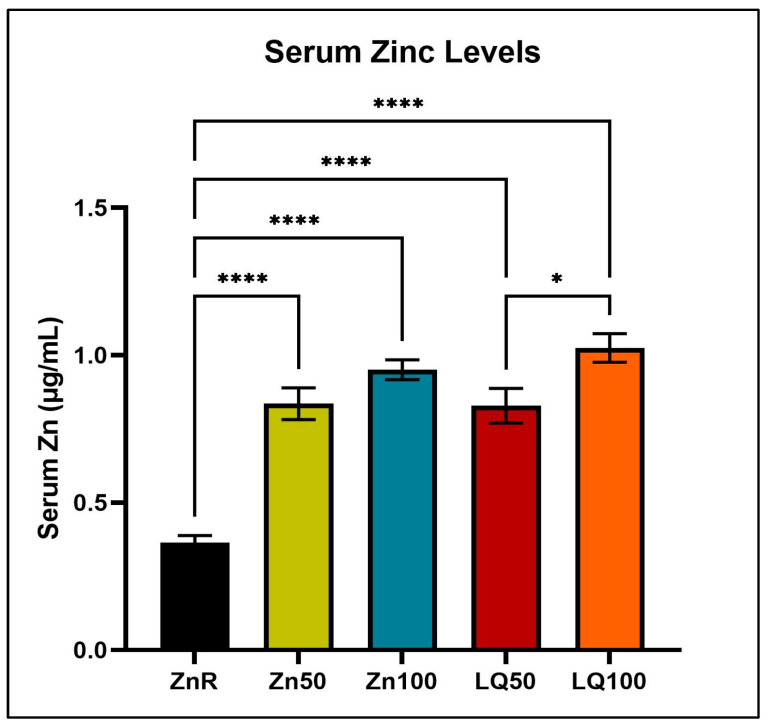
Zinc levels in blood serum of pigs fed Zn restricted (ZnR) or diets supplemented with 50 or 100 ppm of Zn from zinc chloride (Zn50, Zn100, respectively) or 50 and 100 ppm of zinc from an organic source (LQ50 and LQ100, respectively). Data are presented as mean ± SEM. ‘*’ denotes *p* ≤ 0.05; ‘****’ denotes *p* < 0.0001. *n* = 9 for all groups.

**Figure 2 animals-13-02519-f002:**
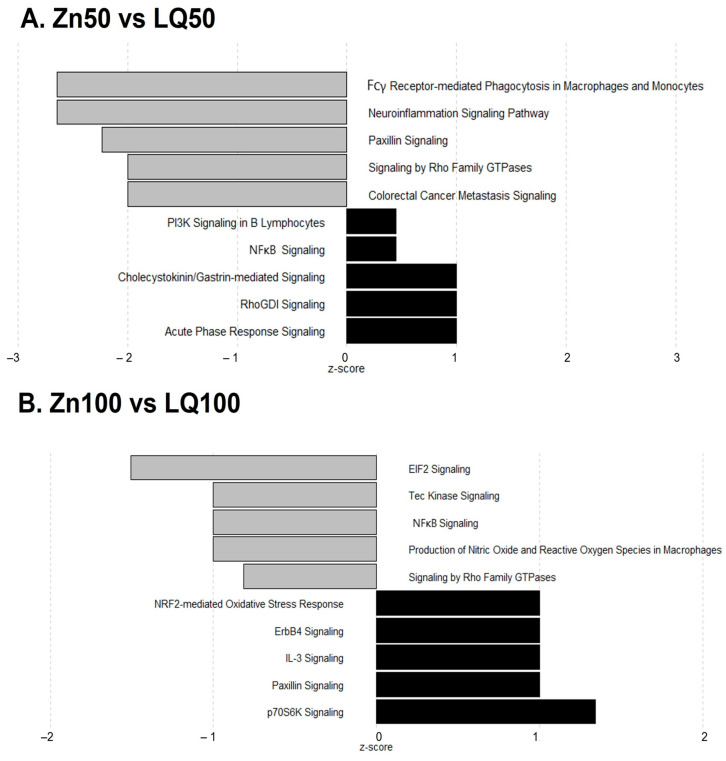
Top five upregulated and downregulated pathways identified through IPA analysis of differentially expressed genes in ileum of pigs fed diets supplemented with 50 ppm (**A**) or 100 ppm (**B**) of Zn from inorganic (Zn50 and Zn100) compared with organic (LQ50 and LQ100) source. Pathways were considered significantly upregulated (black bars) or downregulated (grey bars), respectively, based on z-scores. *n* = 8 for Zn100, Zn50 and LQ100, *n* = 9 for LQ50.

**Figure 3 animals-13-02519-f003:**
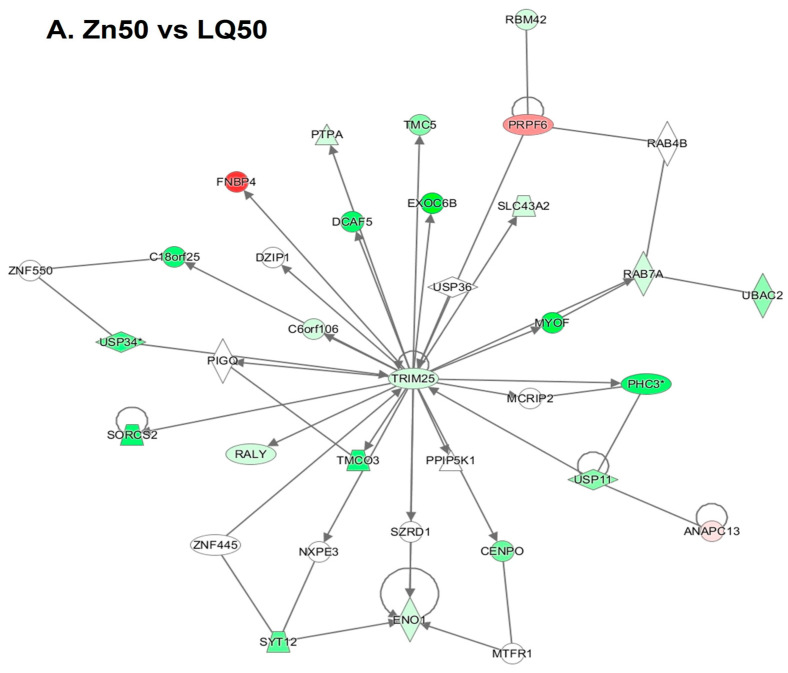
Top functional networks identified in ileum of pigs fed diets supplemented with 50 ppm (**A**) or 100 ppm (**B**) of Zn from inorganic (Zn50 and Zn100) compared with organic (LQ50 and LQ100) source. Genes are denoted as nodes with color representing up- (red) and down- (green) regulated genes. Edges (lines and arrows between nodes) represent direct (solid lines) and indirect (dashed lines) interactions between molecules. *n* = 8 for Zn100, Zn50 and LQ100, *n* = 9 for LQ50. The asterisk in the network means that the gene might have other interactions that are not shown in the current context.

**Figure 4 animals-13-02519-f004:**
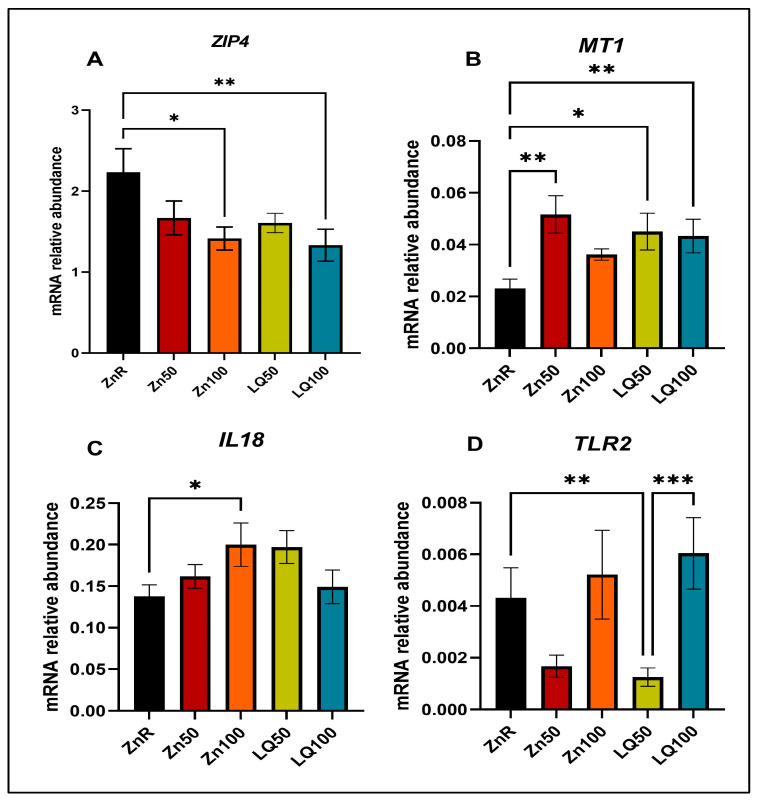
Gene expression in jejunal samples determined by qPCR of (**A**) *ZIP4,* (**B**) *MT1,* (**C**) *IL18,* and (**D**) *TLR2*. Data are presented as mean ± SEM. ‘*’ denotes *p* ≤ 0.05; ‘**’ denotes *p* < 0.01, ‘***’ denotes *p <* 0.001. *n* = 9 for all groups.

**Figure 5 animals-13-02519-f005:**
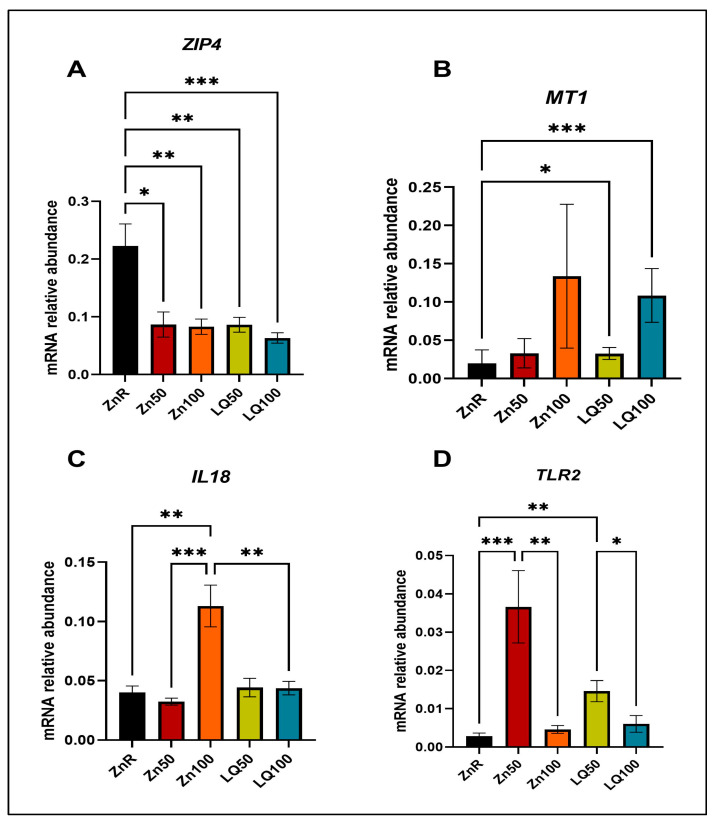
Gene expression in ileal samples determined by qPCR of (**A**) *ZIP4,* (**B**) *MT1,* (**C**) *IL18,* and (**D**) *TLR2*. Data are presented as mean ± SEM. ‘*’ denotes *p* ≤ 0.05; ‘**’ denotes *p* < 0.01, ‘***’ denotes *p* < 0.001. *n* = 7 for ZnR, *n* = 8 for ZnCl100, ZnCl50 and LQ100, *n* = 9 for LQ50.

**Figure 6 animals-13-02519-f006:**
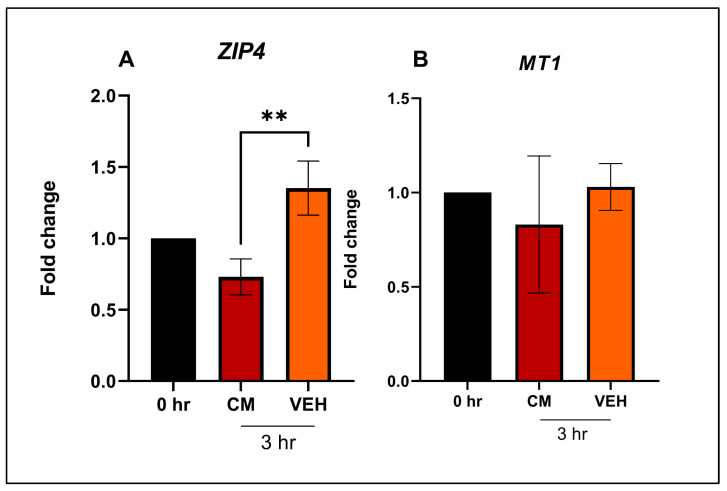
Gene expression of (A) *ZIP4* and (B) *MT1* in 2D swine enteroids. Enteroids were collected at the beginning of the experiment (0 h), or 3 h after culture media (CM) or vehicle (VEH) were added to the luminal side. Data are presented as mean ± SEM. ‘**’ denotes *p* < 0.01. *n* = 3 for all treatments.

**Figure 7 animals-13-02519-f007:**
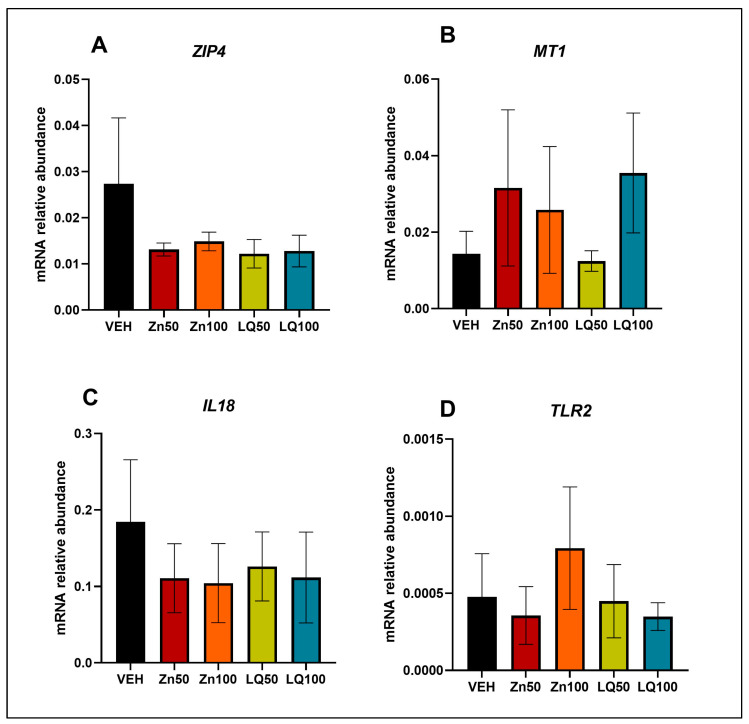
Gene expression of (A) *ZIP4,* (B) *MT1,* (C) *IL18,* and (D) *TLR2* in 2D swine enteroids luminally exposed to Zn-deficient vehicle (VEH) or different sources of Zn. Data are presented as mean ± SEM. *n* = 3.

**Table 1 animals-13-02519-t001:** Ingredient and nutrient composition of the basal diet (as fed basis) ^1^.

Item	Diet Composition
**Ingredients (%)**	
Corn	68.15
Soybean meal	27.50
Soybean oil	1.00
Zn-free vitamin and mineral premix	3.35
Total	100.00
**Calculated (%)**	
Dry matter	85.93
Crude protein	18.74
Metabolizable energy (kcal/kg)	3305
**SID AA (%) ***	
Lys	0.88
Met	0.27
Thr	0.58
Trp	0.20
Leu	1.45
Ile	0.68
Val	0.75
Calcium (%)	0.77
Phosphorus (%)	0.66

Zn-restricted basal diet was supplemented with 50 ppm or 100 ppm of Zn from ZnCl_2_ or LQ-Zn to prepare treatment diets. ^1^ Formulated based on metabolizable energy values for corn and soybean meal. * SID AA: Standardized ileal digestible amino acids.

**Table 2 animals-13-02519-t002:** Calculated nutrient composition of the Zn-free vitamin and mineral premix.

Vitamin/Trace Mineral	Concentration Per kg of Premix
Vitamin A	245,152 I.U. *
Vitamin D3	45,966 I.U.
Vitamin E	919 I.U.
Vitamin K	92 mg
Niacin	919 mg
Pantothenic acid	613 mg
Riboflavin	153 mg
Vitamin B12	919 µg
Copper	107 mg
Iodine	8.2 mg
Iron	917 mg
Manganese	153 mg
Selenium	8.2 mg
Calcium	21.4% of mix
Phosphorus	8.53% of mix
Salt	12.4% of mix

* I.U.: International Unit. Trace mineral sources: Iodine (ethylenediamine dihydroiodide), polysaccharide Fe, Mn, and Cu complexes, and selenium selenite. Dietary treatments were as follows: (a) Zn-restricted diet (ZnR): Basal diet containing Zn-free vitamin-mineral mix; (b) Zn50: ZnR + 50 ppm Zn from ZnCl_2_; (c) Zn100: ZnR+ 100 ppm Zn from ZnCl_2_; (d) LQ50: ZnR + 50 ppm Zn from LQ-Zn; and (e) LQ100: ZnR + 100 ppm Zn from LQ-Zn.

**Table 3 animals-13-02519-t003:** Primer sequences used for gene expression analysis in swine tissue and enteroids.

Gene *	Forward Primer	Reverse Primer
*GAPDH*	ATCCTGGGCTACACTGAGGAC	AAGTGGTCGTTGAGGGCAATG
*IL18*	ACCCGTATCCCCAAGATCCA	TTGCGCTTGATGAGGACAG
*MT1*	GCTGTGCCTGATGTGACGAA	AGGAAGACGCTGGGTTGGT
*TLR2*	TCCCCAGCGTTTCTGTAAGC	ATGAACGCAGCCCAGGACTA
*ZIP4*	CTGCACACACATGATGGGGA	GGTTGAAAAGGCTCTCGAACA

* *GAPDH:* Glyceraldehyde-3-phosphate dehydrogenase, *IL18:* Interleukin 18, *MT1:* Metallothionein1, *TLR2:* Toll like receptor 2, *ZIP4:* Zrt- and Irt-Like Protein 4.

**Table 4 animals-13-02519-t004:** Differentially regulated IPA pathways in Zn50 group compared with LQ50 group.

Pathways	−log (*p*-Value)	z Score	Ratio	Genes
Neuroinflammation Signaling Pathway	1.87 × 10^0^	−2.646	2.24 × 10^−2^	*IL18*, *IRF7*, *NOS2*, *NFATC2*, *TLR2*, *TBK1*, *MAPK3*
Fcγ Receptor-mediated Phagocytosis in Macrophages and Monocytes	5.02 × 10^0^	−2.646	7.61 × 10^−2^	*ACTG1*, *ACTB*, *NCK2*, *FYB1*, *ARF6*, *RAC1*, *MAPK3*
Paxillin Signaling	2.53 × 10^0^	−2.236	4.13 × 10^−2^	*ACTG1*, *ACTB*, *NCK2*, *ARF6*, *RAC1*
Colorectal Cancer Metastasis Signaling	8.24 × 10^−1^	−2.000	1.57 × 10^−2^	*NOS2*, *TLR2*, *RAC1*, *MAPK3*
Signaling by Rho Family GTPases	8.37 × 10^−1^	−2.000	1.59 × 10^−2^	*ACTG1*, *ACTB*, *RAC1*, *MAPK3*
Agrin Interactions at Neuromuscular Junction	2.50 × 10^0^	−2.000	5.33 × 10^−2^	*ACTG1*, *ACTB*, *RAC1*, *MAPK3*
Integrin Signaling	4.11 × 10^0^	−1.414	4.07 × 10^−2^	*TSPAN1*, *ACTG1*, *ACTB*, *CAPN1*, *NCK2*, *ARF6*, *TNK2*, *RAC1*, *MAPK3*
Actin Cytoskeleton Signaling	1.41 × 10^0^	−1.342	2.16 × 10^−2^	*ACTG1*, *ACTB*, *MYH11*, *RAC1*, *MAPK3*
Death Receptor Signaling	3.03 × 10^0^	−1.342	5.38 × 10^−2^	*APAF1*, *ACTG1*, *ACTB*, *PARP9*, *TBK1*
Production of Nitric Oxide and Reactive Oxygen Species in Macrophages	2.97 × 10^0^	−1.134	3.57 × 10^−2^	*NOS2*, *PTPA*, *ARG2*, *APOC3*, *TLR2*, *RAC1*, *MAPK3*
Opioid Signaling Pathway	8.64 × 10^−1^	−1.000	1.63 × 10^−2^	*CLTA*, *RAC1*, *AP1B1*, *MAPK3*
Huntington’s Disease Signaling	1.79 × 10^0^	−1.000	2.38 × 10^−2^	*APAF1*, *CLTA*, *TCERG1*, *PLCB4*, *CAPN1*, *MAPK3*
Role of Pattern Recognition Receptors in Recognition of Bacteria and Viruses	2.27 × 10^0^	−1.000	3.60 × 10^−2^	*IL18*, *IRF7*, *C3*, *TLR2*, *MAPK3*
ILK Signaling	2.96 × 10^0^	−0.816	3.55 × 10^−2^	*NOS2*, *ACTG1*, *ACTB*, *PTPA*, *NCK2*, *MYH11*, *MAPK3*
LXR/RXR Activation	3.35 × 10^0^	−0.816	4.96 × 10^−2^	*IL18*, *NOS2*, *C3*, *ARG2*, *APOC3*, *IL33*
Protein Kinase A Signaling	6.61 × 10^−1^	−0.447	1.25 × 10^−2^	*ANAPC13*, *SMPDL3B*, *NFATC2*, *PLCB4*, *MAPK3*
Dendritic Cell Maturation	1.67 × 10^0^	0.447	2.55 × 10^−2^	*IL18*, *PLCB4*, *TLR2*, *IL33*, *MAPK3*
NF-κB Signaling	1.75 × 10^0^	0.447	2.67 × 10^−2^	*IL18*, *TNIP1*, *TLR2*, *IL33*, *TBK1*
PI3K Signaling in B Lymphocytes	2.32 × 10^0^	0.447	3.70 × 10^−2^	*NFATC2*, *C3*, *PLCB4*, *RAC1*, *MAPK3*
Acute Phase Response Signaling	1.27 × 10^0^	1.000	2.27 × 10^−2^	*IL18*, *C3*, *IL33*, *MAPK3*
RhoGDI Signaling	1.28 × 10^0^	1.000	2.29 × 10^−2^	*ACTG1*, *ACTB*, *ARHGDIA*, *RAC1*
Cholecystokinin/Gastrin-mediated Signaling	1.96 × 10^0^	1.000	3.74 × 10^−2^	*IL18*, *PLCB4*, *IL33*, *MAPK3*

**Table 5 animals-13-02519-t005:** Differentially regulated pathways in Zn100 group compared with LQ100 group.

Pathways	−log (*p*-Value)	z Score	Ratio	Genes
EIF2 Signaling	1.19 × 10^1^	−1.508	8.30 × 10^−2^	*EIF3G*, *RPL32*, *RPL13*, *SOS1*, *RPL26*, *RPS3*, *RALB*, *RPL8*, *ATF4*, *EIF3E*, *RPS9*, *RPL18*, *RPL24*, *RPS11*, *EIF3H*, *EIF3K*, *RPS28*, *PIK3C3*, *RPL10A*
Production of Nitric Oxide and Reactive Oxygen Species in Macrophages	9.06 × 10^−1^	−1.000	2.04 × 10^−2^	*RHOH*, *PRKCD*, *IRF8*, *PIK3C3*
NF-κB Signaling	9.60 × 10^−1^	−1.000	2.14 × 10^−2^	*RALB*, *MAP4K4*, *LCK*, *PIK3C3*
Tec Kinase Signaling	1.07 × 10^0^	−1.000	2.34 × 10^−2^	*RHOH*, *PRKCD*, *LCK*, *PIK3C3*
Signaling by Rho Family GTPases	1.43 × 10^0^	−0.816	2.38 × 10^−2^	*RHOH*, *ARHGEF2*, *ARPC4*, *MYLK*, *PLD1*, *PIK3C3*
Superpathway of Inositol Phosphate Compounds	1.54 × 10^0^	−0.816	2.53 × 10^−2^	*DUSP5*, *IPPK*, *LCK*, *PPM1H*, *PTPRF*, *PIK3C3*
ERK/MAPK Signaling	1.82 × 10^0^	−0.816	2.93 × 10^−2^	*RALB*, *PRKCD*, *ATF4*, *SOS1*, *PIK3C3*, *PLA2G2A*
Colorectal Cancer Metastasis Signaling	9.85 × 10^−1^	−0.447	1.96 × 10^−2^	*RALB*, *RHOH*, *TCF3*, *SOS1*, *PIK3C3*
3-phosphoinositide Biosynthesis	1.32 × 10^0^	−0.447	2.48 × 10^−2^	*DUSP5*, *LCK*, *PPM1H*, *PTPRF*, *PIK3C3*
PPARα/RXRα Activation	1.45 × 10^0^	−0.447	2.69 × 10^−2^	*RALB*, *MAP4K4*, *ACOX1*, *SOS1*, *GPD2*
Gαq Signaling	1.67 × 10^0^	−0.447	3.09 × 10^−2^	*RHOH*, *PRKCD*, *PLD3*, *PLD1*, *PIK3C3*
ILK Signaling	1.89 × 10^0^	−0.447	3.05 × 10^−2^	*RHOH*, *MYH14*, *ATF4*, *FBLIM1*, *PARVA*, *PIK3C3*
SAPK/JNK Signaling	2.32 × 10^0^	−0.447	4.46 × 10^−2^	*RALB*, *MAP4K4*, *LCK*, *SOS1*, *PIK3C3*
IL-8 Signaling	2.41 × 10^0^	−0.378	3.43 × 10^−2^	*RALB*, *RHOH*, *PRKCD*, *MAP4K4*, *PLD3*, *PLD1*, *PIK3C3*
Endothelin-1 Signaling	2.47 × 10^0^	0.378	3.52 × 10^−2^	*RALB*, *PRKCD*, *PLD3*, *PLD1*, *SOS1*, *PIK3C3*, *PLA2G2A*
Integrin Signaling	2.83 × 10^0^	0.378	3.62 × 10^−2^	*RALB*, *RHOH*, *ARPC4*, *MYLK*, *TSPAN5*, *PARVA*, *SOS1*, *PIK3C3*
Opioid Signaling Pathway	1.03 × 10^0^	0.447	2.03 × 10^−2^	*RALB*, *PRKCD*, *ATF4*, *LCK*, *SOS1*
Fc Epsilon RI Signaling	2.09 × 10^0^	0.447	3.94 × 10^−2^	*RALB*, *PRKCD*, *SOS1*, *PIK3C3*, *PLA2G2A*
NGF Signaling	2.12 × 10^0^	0.447	4.00 × 10^−2^	*RALB*, *PRKCD*, *ATF4*, *SOS1*, *PIK3C3*
VEGF Family Ligand–Receptor Interactions	2.61 × 10^0^	0.447	5.21 × 10^−2^	*RALB*, *PRKCD*, *SOS1*, *PIK3C3*, *PLA2G2A*
Glioblastoma Multiforme Signaling	2.19 × 10^0^	0.816	3.53 × 10^−2^	*RALB*, *RHOH*, *PRKCD*, *TCF3*, *SOS1*, *PIK3C3*
Thrombin Signaling	2.33 × 10^0^	0.816	3.32 × 10^−2^	*RALB*, *RHOH*, *PRKCD*, *ARHGEF2*, *MYLK*, *SOS1*, *PIK3C3*
GNRH Signaling	1.07 × 10^0^	1.000	2.35 × 10^−2^	*RALB*, *PRKCD*, *ATF4*, *SOS1*
Role of NFAT in Cardiac Hypertrophy	1.15 × 10^0^	1.000	2.21 × 10^−2^	*RALB*, *PRKCD*, *SLC8A1*, *SOS1*, *PIK3C3*
Insulin Receptor Signaling	1.26 × 10^0^	1.000	2.72 × 10^−2^	*RALB*, *PTPRF*, *SOS1*, *PIK3C3*
Renin-Angiotensin Signaling	1.42 × 10^0^	1.000	3.08 × 10^−2^	*RALB*, *PARVA*, *SOS1*, *PIK3C3*
Glioma Signaling	1.50 × 10^0^	1.000	3.28 × 10^−2^	*RALB*, *PARVA*, *SOS1*, *PIK3C3*
HGF Signaling	1.52 × 10^0^	1.000	3.31 × 10^−2^	*RALB*, *PARVA*, *SOS1*, *PIK3C3*
Paxillin Signaling	1.52 × 10^0^	1.000	3.31 × 10^−2^	*RALB*, *PARVA*, *SOS1*, *PIK3C3*
Cholecystokinin/Gastrin-mediated Signaling	1.69 × 10^0^	1.000	3.74 × 10^−2^	*RALB*, *RHOH*, *PRKCD*, *SOS1*
ErbB Signaling	1.70 × 10^0^	1.000	3.77 × 10^−2^	*RALB*, *PRKCD*, *SOS1*, *PIK3C3*
Prolactin Signaling	1.92 × 10^0^	1.000	4.40 × 10^−2^	*RALB*, *PRKCD*, *SOS1*, *PIK3C3*
IL-3 Signaling	1.94 × 10^0^	1.000	4.44 × 10^−2^	*RALB*, *PRKCD*, *SOS1*, *PIK3C3*
ErbB4 Signaling	2.11 × 10^0^	1.000	5.00 × 10^−2^	*RALB*, *PRKCD*, *SOS1*, *PIK3C3*
Thrombopoietin Signaling	2.27 × 10^0^	1.000	5.56 × 10^−2^	*RALB*, *PRKCD*, *SOS1*, *PIK3C3*
NRF2-mediated Oxidative Stress Response	2.43 × 10^0^	1.000	3.47 × 10^−2^	*RALB*, *DNAJC6*, *PRKCD*, *ATF4*, *GSR*, *MGST3*, *PIK3C3*
p70S6K Signaling	1.92 × 10^0^	1.342	3.57 × 10^−2^	*RALB*, *PRKCD*, *PLD1*, *SOS1*, *PIK3C3*

## Data Availability

The raw data and processed data have been deposited in the NCBI Gene Expression Omnibus (GEO) under accession number GSE181343.

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
