# Peer review of "Dietary Zinc Supplemented in Organic Form Affects the Expression of Inflammatory Molecules in Swine Intestine"

_animals, 2023, doi:10.3390/ani13152519_

Round 1

Reviewer 1 Report

Zinc is vital to the gut health of pigs, high dosage ZnO has been widely used to control the diarrhea incidence of weaned piglets. Plenty of studies have been conducted to evaluate different sources of zinc on gut health of pigs. The authors identified intestinal epithelial responses to organic and inorganic Zn via in vivo and in vitro studies. The authors observed organic Zn might improve immune response through TLR2 signaling while decrease expression of proinflammatory cytokines in non-epithelial cells of the intestinal mucosa, which could give very useful information to feed industry and swine nutrition. However, the experimental design and animals used in this study are major issues to consider the manuscript as the acceptable paper.

1. The authors should provide the reasons why choose ZnCl2 instead of ZnO.

2. The experimental design in the in vivo study was improperly. All the nine pigs in each treatment were fed in one pen, thus it rather difficult to calculate the growth performance due to the feed intake. 

3. The level of crude protein in this study is much higher the NRC requirement for finisher pigs with 77.5 kg.

4. The first part (Lines 314-322) in the 4.4 section should be moved to M&M section instead of Results section.

5. The discussion mainly focused on the results concerning gene expression, while limited information of RNAseq was discussed. Thus, the section should be enriched.

6. Section “Methods” should be changed as “Materials and Methods”

Author Response

Response to Reviewer 1:

  1. The authors should provide the reasons why choose ZnCl2instead of ZnO.

Response: Zinc is absorbed as Zn2+ ions in the intestine irrespective of the zinc salt it is sourced from. The most common inorganic salt used for feeding pigs is ZnO. However, ZnO dissociates in the gastric environment because of the pH and gastric acid, thus converting into ZnCl2 when reaching the intestine. Moreover, zinc chloride has higher bioavailability than zinc oxide. To avoid confounding results because of variability of zinc oxide / chloride conversion and bioavailability as consequence, we decided to use zinc chloride as the inorganic source of zinc.

  1. The experimental design in the in vivostudy was improperly. All the nine pigs in each treatment were fed in one pen, thus it rather difficult to calculate the growth performance due to the feed intake.

Response: We agree that assessing growth performance is not possible in this study. For the animal study, pen is the experimental unit. Animals and experimental diets were randomly allocated to each pen. Although we cannot calculate average daily feed intake of each animal, and performance, we can calculate the average daily gain for each animal. We do not aim to make statements on growth performance in this study. We used weight gain because that is an important indicator of zinc deficiency. We revised the wording along the paper to indicate that we are not evaluating performance, but only weight gain.

  1. The level of crude protein in this study is much higher the NRC requirement for finisher pigs with 77.5 kg.

Response: Pigs require 16% crude protein as per NRC recommendations. Our basal diet provided 18.74% crude protein which meets the NRC requirements. The diet used is similar to what has been reported as standard diet in numerous publications. If the level of protein is considered too high, it is important to remember that all pigs received that same amount of protein, thus any potential changes due to protein levels would be equal among groups.

  1. The first part (Lines 314-322) in the 4.4 section should be moved to M&M section instead of Results section.

Response: Edited accordingly in lines 116-117.

  1. The discussion mainly focused on the results concerning gene expression, while limited information of RNAseq was discussed. Thus, the section should be enriched.

Response: The current study aimed to explore the differences between organic and inorganic zinc sources via RNA-seq analysis. In a previous report (reference 21) we have reported and discussed findings related to Zn restriction and supplementation using the same animal model. This paper is focused on the differences among zinc sources and exploring further the potential role of zinc on TLR2 and IL18 expressions. Following the reviewer’s suggestion, we edited the discussion section by including a brief discussion on the findings of network analysis in lines 373-393, which add to what is already in the literature from our study in reference 21.

  1. Section “Methods” should be changed as “Materials and Methods”

Response: Edited as suggested.

Reviewer 2 Report

Dear authors,

This is an interesting study. The simple summary and the abstract have to be revised.  Some precision is needed in the methodology section. In the discussion section, there is no discussion on RNA Sequencing and Differential Expression results, which should be the core of this study. When discussing results of gene expression, the authors have to be careful not to extrapolate results obtained with 2 genes to whole metabolic pathways or systems. Although the two analyzed genes gave some support for data from RNA sequencing and differential expression analysis, one would expect more genes to be analyzed so results would be more solid. 

Please, find comments suggestions and question in the enclosed PDF file.

Author Response

Response to Reviewer 2:

Simple summary

  1. I suggest the authors to remove this part.

Response: Edited as suggested.

  1. Please, specify the type of biological function these genes were related to (antioxidant system, immune system, nutrient transporters, etc).

Additionally, clarify if these gene expression analyses were performed in the ileum mucosa or other part of the ileum tissue.

Response: The simple summary has been edited. The RNA was extracted from whole ileum pieces including most layers, so we are keeping “ileum” to avoid the interpretation that we collected mucosal scrapings or somewhat separating intestinal layers. Because of the mixed nature of the sample, we used the in vitro system to identify if epithelial cells would be the drivers of these changes, as those are the cells that are directly exposed to zinc sources.

  1. Considering it is a simple summary, I would not use these technical terms but simply state that Organic Zn lowered inflammation compared to inorganic Zn.

Response: We did not assess changes in inflammation, we evaluated changes in gene expression, so we kept the wording to avoid overstating the findings.

  1. There was nothing in the previous lines concerning performance nor the use of an experimental model of stress. I suggest the authors to better introduce the experimental conditions presenting the different treatments, the stress model, and the most important analyses.

Response:  The study did not include any stress model, so we revised the summary to indicate changes in gene expression of immune related molecules, and not performance or whole animal responses.

Abstract

  1. This is a very broad affirmative. What kind of challenges are the authors making reference to? Are they heat stress challenges? Are they gut infection challenges? Please, be more specific.

Response: We agree this is a broad statement, but it summarizes a number of reports on different species and challenges. Those reports are summarized in the introducing, lines 44 to 52.

  1. Again, be more specific on the type of responses the authors were evaluating. Immune response, antioxidant response, microbiological response, epigenetic response, gene expression response...

Response: As our approach was to use RNAseq, which is an exploratory tool, we were not evaluating specific responses but aiming to identify which responses may be affected by zinc sources.

Please, indicate weight and/or age.

Response: Information added.

  1. There is no description of the evaluated parameters nor the statistical analysis. Was it a one-way analysis with 5 treatments or was it a 2x2 factorial plus control?

Response: It was a differential gene expression analysis (RNAseq) with pathway analysis.

  1. mucosa?

Response: We used whole tissue.

  1. This is too vague. What are these expected changes? Please, add results with P value.

Response: Added.

  1. In which tissue? Both? Please, add P value.

Response: Ileum, values added.

  1. The results below state the presence of the Zn-restricted diet as well. Please, correct.

Response: Abstract lines 31-36 edited in response to reviewer’s comments.

  1. Results

Response: Abstract was edited in response to reviewer’s comments.

  1. Based on the present results, the authors can conclude that organic Zn may stimulate the TLR2 signaling but not the immune system as a whole.

Response: We only measured changes in gene expression so we can’t make statements on signaling pathway activation or active immune response changes.

  1. The present study evaluated only 1 proinflammatory cytokine. Therefore, it has to conclude based on the one cytokine analysed and cannot make inferences on other cytokines.

Response: Yes. We reworded the abstract to reflect these comments.

Introduction

  1. Maybe it would be more prudent to state "mammals". If the authors want to stick with their original text, please, provide a reference.

Response: Changed as suggested and references added.

Methods

  1. Please, indicate the concentration of Zn in the diets.

Response: Concentration of Zn in diets is indicated in lines 108-113 and table 2 footnote.

  1. If pigs had an initial body weight of 77 kg, why not using recommendations for pigs weighing 75-100 kg?

Response: For finisher pigs, the recommendation is 50 ppm, which we used in the study.

  1. Why did the authors decided to use ZnCl2 as the inorganic source?

Response: Zinc is absorbed as Zn2+ ions in the intestine irrespective of the zinc salt it is sourced from. The most common inorganic salt used for feeding pigs is ZnO. However, ZnO dissociates in the gastric environment because of the pH and gastric acid, thus converting into ZnCl2 when reaching the intestine. Moreover, zinc chloride has higher bioavailability than zinc oxide. To avoid confounding results because of variability of zinc oxide / chloride conversion and bioavailability as consequence, we decided to use zinc chloride.

Please, indicate what type of organic source it is (protein-Zn complex; amino acid-Zn complex, etc).

Response: Organic zinc source is a commercially available zinc-amino acid complex. The information was added in line 93.

  1. Was there any waiting time for clot to form before centrifugation

Response: Blood samples were allowed to clot for 20 minutes before centrifugation. Methods part edited accordingly in line 116.

  1. Were they mucosa samples or whole tissue samples?

Response: The samples were whole tissue samples. Methods section edited accordingly in lines 124-125.

  1. Considering that Zn absorption occurs mainly in the upper duodenal tissue, why choosing ileum to perform analysis?

Response: Ileum would be the ideal site to observe changes related to immune function mainly because of the localization of Peyer’s path around the ileal areal when compared to the upper small intestinal regions. As most differences observed in animals fed different sources of zinc are related to response to stress, we figured the ileum provided more information on stress responses than the proximal small intestine.

  1. Why did the authors decide to choose such precise genes (IL18 and TLR2) instead of using a larger set of genes (that could have shown the bigger picture of what Zn effects)? There were many genes that could have been chosen for qPCR confirmation. I believe the main objective here was to confirm findings from the RNA Sequencing and Differential Expression Analysis, but even there, only 2 genes seems not to be enough.

Response: We focused on TLR2 and IL18 because they were the common factor of the pathways identified in the analysis (tables 4 and 5). Although we focused on these two to validate the RNAseq results and with the expectation they would be major responders to zinc sources, there are many other genes identified in our study that deserve further analysis.

  1. Was not it used for samples from the in vivo study as well?

Response: Yes, we appreciate the reviewer noticing this omission. The title of table 3 was edited accordingly.

  1. Why 10 % CO2 if the previous test used 5 % CO2?

Response: This was a mistake. Enteroids were incubated at 5% CO2. Line 198 edited accordingly.

Data analyses

  1. I suggest the authors to use a 2x2 factorial design (2 sources and 2 levels) plus a comparison of control vs supplemented animals. I believe it would make the presentation and discussion of results clearer.

Response: We did not use a factorial design because we are interested in the comparison between similar levels of zinc between both the sources and with the control Zn restricted group. Also, because there is one pen per diet, there is not enough sample to evaluate animal performance and apply a factorial design. Reviewer 1 commented on that and in clarifying this point (that we can’t assess animal performance, just changes in daily gain) and the report was revised to indicate this.

Results

  1. P value?

Response: P-value added in line 217.

  1. ileum?

Response: Edited accordingly (line 235).

  1. This information should be added to the methodology section.

Response: Edited accordingly in lines 124-125.

Discussion

  1. There is basically no discussion on RNA Sequencing and Differential Expression Analysis, which should be the core of this study.

Response: The current study aimed to explore the differences between organic and inorganic zinc sources via RNA-seq analysis. In a previous report (reference 21) we have reported and discussed findings related to Zn restriction and supplementation using the same animal model. This paper is focused on the differences among zinc sources and exploring further the potential role of zinc on TLR2 and IL18 Following the reviewer’s suggestion, we edited the discussion section by including a brief discussion on the findings of network analysis in lines 373-393, which add to what is already in the literature from our study in reference 21.

29. I appreciate the care the authors took by discussing these results as "suggesting a possibility". Although the potential modulation of TLR2 signaling may indeed be a possibility, I believe it is not appropriate to state that the synthesis of "proinflammatory cutokines" will be possibly reduced only because one proinflammatiry cytokine was reduced. It would take some more cytokines to support this statement. The same for the improved epithelial barrier. Again, although the authors carefully approached this possibility, I believe they do not have enough data to extrapolate on that matter.

Response: We agree with the reviewer. We hope the data generated by our study will provide the steppingstone for more research leading to understand the role of zinc supplementation and zinc sources in animal health. We added text in lines 389-392 to reflect this need for further studies.

Round 2

Reviewer 2 Report

Dear authors,

All comments/questions were satisfactorily answered.

Thank you